# Selection of Immature Cat Oocytes with Brilliant Cresyl Blue Stain Improves In Vitro Embryo Production during Non-Breeding Season

**DOI:** 10.3390/ani10091496

**Published:** 2020-08-24

**Authors:** Anna Rita Piras, Federica Ariu, Maria-Teresa Zedda, Maria-Teresa Paramio, Luisa Bogliolo

**Affiliations:** 1Department of Veterinary Medicine, University of Sassari, 070100 Sassari, Italy; Federica@uniss.it (F.A.); Zedda@uniss.it (M.-T.Z.); Luis@uniss.it (L.B.); 2Departament de ciencia Animal i Dels Aliments, Universitat Autònoma de Barcelona, Bellaterra, 08193 Barcelona, Spain; teresa.paramio@uab.es

**Keywords:** BCB, blastocyst, domestic cat, in vitro maturation, mitochondria, ROS, seasonality

## Abstract

**Simple Summary:**

The domestic cat is commonly used as a model for the development of assisted reproductive technologies, including in vitro embryo production (IVEP) in felid species. Seasonal reproduction is a feature of domestic cats as well as of several species of wild feline. Likewise, the number and the quality of blastocysts produced in in vitro systems is linked to season. Maintaining stable in vitro embryo production throughout the year is crucial not only for research purposes but also for programs aimed at protecting endangered felines. We assess whether using Brilliant Cresyl Blue (BCB) selection in addition to the classical morphological selection could improve the IVEP outcomes during non-breeding season. Blastocyst yield and quality of the embryos (hatching rate and blastocyst cell numbers) were higher after IVM/IVF in oocytes defined as BCB+ (colored cytoplasm) based on the BCB test than in oocytes only morphologically selected. Furthermore, no adverse effects on bioenergetic/oxidative status were observed in oocytes subjected to BCB staining. In conclusion, BCB test implementation in IVEP programs might ensure a steady output of domestic cat blastocysts throughout the year.

**Abstract:**

In domestic cats, the maturation, fertilization, and development potential in vitro decreases during the non-breeding season. This study aims at evaluating the efficacy of Brilliant Cresyl Blue (BCB) staining in selecting developmentally competent oocytes to be used in in vitro embryo production (IVEP) programs in order to overcome the season variability in blastocyst yield. Cumulus-oocytes complexes (COCs) collected from antral follicles of domestic cat ovaries during the anestrus phase (July to November) were selected by BCB staining and classified as BCB+ (colored cytoplasm) and BCB− (colorless cytoplasm). COCs not exposed to BCB staining were used as control. Before and after in vitro maturation mitochondrial activity and reactive oxygen species (ROS) were measured. Following in vitro fertilization, blastocyst rate, hatching rate, and blastocyst cell numbers were recorded. The results show that BCB staining did not alter the mitochondrial function and ROS production in cat oocytes. BCB+ oocytes presented a higher (*p* < 0.05) blastocyst rate, hatching rate, and blastocyst cell number than BCB− and control oocytes. In conclusion, BCB staining does not affect the bioenergetic/oxidative status of the oocyte while being a useful tool for selecting good quality oocytes to increase IVEP in domestic cats during non-breeding season.

## 1. Introduction

Over the last decades, appreciable advances have been made in the development of assisted reproductive technology (ART) for feline embryos in vitro production. The possibility of using domestic cat gametes as a valuable model has allowed the improvement of in vitro embryo production (IVEP) technology that might be helpful for preserving endangered wild felids [1]. Although living offspring have been produced after the transfer of in vitro produced embryos both in domestic [2,3] and non-domestic cats [4], the efficiency of IVEP is still low compared to the level achieved in farm animals [5].

Among the multitude of factors which may affect the developmental potential of cat oocytes following in vitro maturation (IVM) and fertilization (IVF) [6,7,8,9], the influence of season and ovarian status has previously been highlighted [10,11,12]. Under natural conditions, the free-ranging domestic cat is a seasonally polyestrous long-day species. In the northern hemisphere, the breeding season begins in January or February, with the highest incidence of estrus in February and March, and ends between June and November [13]. Spindler and Wildt [10] first reported that the time of the year has a different impact on the quality of domestic cats oocytes. In vitro embryo development up to the blastocyst stage after IVM\IVF of domestic cat oocytes is significantly reduced during the non-breeding season compared to the reproductive season [10,11,12]. The selection of immature cumulus oocyte complexes (COCs) for IVM in cat, as well as in other species, is routinely based on the microscopical assessment of morphological features [9] such as cumulus thickness, ooplasm compactness and homogeneity, and oocytes sizes [9]. However, evidence shows that the morphological criteria do not really reflect the oocyte ability to develop. Consistently, morphological high-quality oocytes retrieved from domestic cats during the non-breeding season displayed a reduced ability to reach nuclear maturation, to be fertilized, and to develop to the blastocyst stage in vitro [10]. Understanding the mechanism that induces a reduction of oocyte quality during the non-breeding season is still limited. It has been suggested that seasonal variation in follicle stimulating hormone (FSH) levels and receptors [12,14,15] or the lack of antioxidant capacity of the cumulus-oocyte complex might be responsible for reduced oocyte developmental competence [12].

Brilliant Cresyl Blue (BCB) staining is a non-invasive method that has been used in different animal species [16], including in humans [17], for identifying the quality of oocytes retrieved from antral follicles [18]. Several studies in different animal species demonstrated that in vitro maturation, cleavage, and development to blastocyst after IVF were significantly higher in BCB+ selected oocytes [18,19,20,21,22,23].

Similar findings have recently been obtained for feline oocytes [24]. BCB staining allowed selecting domestic cat oocytes likely to be developmentally competent [24]. Indeed, domestic cat oocytes with low G6PDH activity respond to IVM\IVF with high nuclear maturation and development to the morula stage [24]. However, in the abovementioned study, the oocytes were collected from cat ovaries independently of the time of the year, and the resulting data were not separated by months or seasons. Despite the overall positive results obtained with BCB selection, the utility of this test is still controversial. Some studies reported no significant differences in developmental competence between BCB+ and unstained control oocytes [25,26]. Recent evidences do not recommended using this test for oocyte screening before IVM, as BCB is toxic and has a deleterious effect on oocytes [27,28,29].

This study was conceived to test whether the selection of immature oocytes retrieved from domestic cat ovaries during the non-breeding season, by using a combination of COC morphology and BCB staining, would be helpful in selecting developmentally competent oocytes, thus enhancing the in vitro fertilization outcome. The possibility of classifying oocytes according to their developmental potential is particularly important to understand oocyte biology and might help to reduce the negative impact of season on in vitro embryo production especially in the case of wild felines that die unexpectedly during the anestrus period.

## 2. Materials and Methods

All chemical reagents were purchased from Sigma-Aldrich Chemical Co. (St. Louis, MO, USA) unless otherwise specified.

### 2.1. Animal Selection

Healthy, free-ranging queens (Felis catus ≥ 1-year-old) were used in this research. The study was carried out between July and November (non-breeding season) in Sardinia, Italy (40°43′33″ N; 8°33′19″ E). Anestrus was confirmed by visualization of morphological characteristics of the ovaries after ovariohysterectomy. Only inactive ovaries (having neither follicular or luteal activity [10] were included in the study).

Live animals were not used in this study. The samples (ovaries) were collected during routine ovariectomies, following the basic criteria of good veterinary surgery practices.

### 2.2. Cumulus-Oocyte Complexes Collection and Morphological Selection

Ovaries were recovered from ovariohysterectomies carried out at the Veterinary Teaching Hospital Sassari University (Sassari, Italy). Ovaries were placed in sterile 15 mL tubes containing PBS (Phosphate Buffered Saline, Life Technologies, Carlsbad, CA, USA) with penicillin (100 mg/L), streptomycin (100 mg/L) and immediately transported to the laboratory at room temperature (22–25 °C).

Cumulus-oocyte complexes (COCs) were recovered from the antral follicles by slicing the ovarian surface in 25 mM Hepes-buffered TCM 199 supplemented with 0.1% (*w/v*) polyvinyl alcohol (PVA) and antibiotics. The immature COCs were morphologically classified under a stereomicroscope, as previously described by Wood and Wildt [9]. Only COCs with two or more granulosa cell layers surrounding the oocyte and with homogeneous dark cytoplasm were selected.

Selected COCs were randomized in two groups: (1) oocytes not exposed to BCB staining and directly submitted to IVM (control group); (2) oocytes submitted to the BCB test (BCB group).

### 2.3. Brilliant Cresyl Blue (BCB) Staining

The BCB group included COCs washed three times in modified PBS (mPBS: PBS supplemented with 1 g/L glucose, 36 mg/L sodium pyruvates, 0.5 g/L bovine serum albumin (BSA) and 0.05 g/L gentamicin) and incubated with 13 µM of BCB diluted in mPBS for 45 min at 38.5 °C in a humidified air atmosphere [22]. Following BCB incubation, COCs were examined and sorted under a stereomicroscope (Olympus IX 40, Segrate, Italy) according to oocytes cytoplasm coloration (Figure 1) into two groups BCB+ (colored cytoplasm) and BCB− (colorless cytoplasm). COCs were washed three times in mPBS before IVM.

### 2.4. Experimental Design

Firstly, we evaluated the effect of BCB staining on: (i) meiotic competence of BCB and control oocytes and (ii) mitochondrial activity and the level of intracellular reactive oxygen species (ROS) of the oocytes immediately after BCB selection and after in vitro maturation (Experiment 1).

This analysis explored whether dye exposure can have potentially damaging effects on meiotic progression and on oocytes bioenergetic/oxidative status. A previous study [29] reported that BCB staining negatively affects the redox state in porcine oocytes.

In Experiment 2, the developmental competence of BCB+ and BCB−, and control oocytes subjected to IVM\IVF were evaluated. Embryonic cleavage, blastocyst formation and hatching and blastocyst cell number were recorded.

### 2.5. Measurement of Mitochondria Activity and Reactive Oxygen Species (ROS) Levels

Mitochondrial membrane potential and intracellular ROS levels of oocytes were measured through staining with MitoTracker Orange CMTMRos (Molecular Probes, Eugene, OR, USA) and 2′,7′-dichlorodihydrofluorescein diacetate (H_2_DCF-DA, Molecular Probe, Eugene, OR, USA), respectively.

Briefly, denuded immature and mature (selected on the basis of polar body extrusion) oocytes were washed three times in PBS with 0.1% BSA and were incubated for 30 min in PBS with 3% BSA containing 200 nM MitoTracker Orange CMTMRos at 38.5 °C under 5% CO_2_. After the first incubation, the oocytes were washed again in PBS with 0.1% BSA and incubated for 30 min in PBS with 0.1% BSA and 10 μM H_2_DCFDA [30]. After washing in PBS with 0.1% BSA, the oocytes were fixed for 60 min at 38 °C in 2% paraformaldehyde and stained for 5 min with 1 μM Hoechst 33,342 solution. All procedures were carried out while paying attention to reducing the exposure of the sample to light in order to avoid photo-bleaching. The oocytes were placed in Eppendorf tubes containing 1 mL of PBS with 0.1% PVA at 4 °C in the dark until their analysis. The oocytes were transferred to a glass slide into a 5 μL drop of PBS and glycerol (3:1 *v*/*v*) solution, covered with a coverslip, supported by four droplets of Vaseline and sealed with nail varnish.

Mitochondria activity and ROS level measurement were performed by a confocal laser-scanning fluorescence microscope (Leica TCS SP5 CLSM), equipped with 543 nm He/Ne, 488 nm Argon and 405 nm diode lasers and recorded on a host computer.

For mitochondrial evaluation, samples were observed with a helium/neon laser ray at 543 nm to detect MitoTracker Orange CMTMRos (excitation 554 nm). The fluorescence signal of DCF, indicating the ROS level, was measured with an argon-ion laser ray at 488 nm (excitation 460 nm). Images were taken at equatorial plane with 40x magnification under mineral oil. The setting of scanning conditions with respect to laser energy, signal detection, offset, filters and pinhole size were maintained constant for all image acquisitions [30]. Acquisition and image analysis were performed by using Leica LAS lite 170 Image software. In each individual oocyte, a circle region of interest (ROI) delimiting the cytoplasm area of the oocyte was selected and fluorescence intensity was measured and reduced by compensation for the background fluorescence. Intensities of fluorescence were normalized to those of control value.

### 2.6. In Vitro Maturation and Assessment of Nuclear Maturation

Cumulus-oocyte complexes from control and BCB groups were in vitro matured in NaHCO_3_-TCM 199 [31] supplemented with 0.36 mM pyruvate, 2 mM glutamine, 2.2 mM calcium lactate, 1.2 mM cysteine, 4 mg/mL BSA and FSH 1 IU⁄mL and luteinizing hormone (LH) 1 IU/mL in groups of 25–35 in 650 μL of IVM medium, in a humidified atmosphere of 5% CO_2_, at 38.5 °C for 24 h.

At the end of IVM, COCs were completely denuded of granulosa cells via gentle pipetting with a fine bore glass pipette and stained with a solution of Hoechst 33,342 (10 μg/mL) in 1:1 (*v*/*v*) glycerol/PBS [7]. The nuclear configuration was classified under an epifluorescent microscope (Olympus IX 70, Segrate, Italy) as germinal vesicle (GV), germinal vesicle breakdown (GVBD), metaphase I (MI), or metaphase II (MII). Oocytes looking diffusely stained through cytoplasm and with unidentifiable or invisible chromatin were considered degenerated [7].

### 2.7. In Vitro Fertilization and Embryo Culture

Spermatozoa were collected from epididymides of adult cats [32] following routine orchiectomy and frozen according to the procedure described by Tsutsui et al. [33]. Frozen-thawed spermatozoa were selected by the swim-up technique and used at a final concentration of 5 × 10^5^ sperm/mL for IVF.

After in vitro maturation, COCs were partially denuded by gentle pipetting and washed in synthetic oviductal fluid (SOF) containing 6 mg/mL BSA, 100 IU/mL penicillin, 50 μg/mL gentamicin [31]. COCs and spermatozoa were co-incubated in four-wells Petri dishes with 450 μL of SOF covered with mineral oil, for approximately 22 h at 38.5 °C in a humidified atmosphere of 5% CO_2_ in air [31]. Presumptive zygotes (day 1 post IVF) were washed and cultured in 650 μL of SOF containing 4 mg/mL BSA, and 100 IU/mL penicillin and 1% MEM non-essential amino acids (IVC-1 medium). On day 3, the embryos were transferred to IVC-2 medium (SOF supplemented with 10% fetal calf serum (FCS) and 2% MEM essential amino acids (EAA), and cultured at 38.5 °C in a humidified atmosphere of 5% CO_2_ in air for additional 4 days [31]. Embryonic cleavage was assessed at 24 h post-fertilization, blastocyst formation and hatching at 7 days post-fertilization.

### 2.8. Assessment of Blastocyst Cell Number

The blastocyst cell number was evaluated by differential staining of the inner cell mass (ICM) and trophectoderm (TE) cell compartments [34]. Briefly, seven-day-old blastocysts were exposed to 1% Triton X-100 in 20 mM HEPES-buffered TCM 199 containing 30 mg/mL propidium iodide (PI) for 35–40 s. The blastocysts were then transferred into ice-cold ethanol for 2–5 s. Finally, the blastocysts were incubated in medium with 50% (*v*/*v*) glycerol and ethanol containing 0.1 mg/mL bis-benzimide (Hoechst 33,342) for 5 min. Embryos were subsequently transferred into a small droplet of glycerol on a glass slide and examined under epifluorescent microscope (Olympus IX70, Segrate, Italy). A digital image of each embryo was taken, and the numbers of ICM (blue) and TE (red) nuclei were counted by the microscope system (Image J 1.5Oi).

### 2.9. Statistical Analysis

All statistical analyses were performed using STATA\IC 11.28 (StataCorp LP, Lakeway Drive, college station, TX, USA). Data on nuclear maturation, cleavage, and development to blastocyst stage were analyzed by chi-square test. The overall chi-square was calculated and found to be significant before the Fisher’s exact test was conducted to identify differences among the experimental classes.

The Shapiro-Wilk test [35] was used to verify the normal distribution of data on blastocyst cell number, intracellular ROS level and mitochondria activity. Data (mean ± SEM) for blastocyst cell numbers were normally distributed and were analyzed using one-way ANOVA followed by Bonferroni correction as post-hoc test. Data (mean ± SEM) for intracellular ROS level and mitochondrial activity were not normally distributed and were analyzed with a non-parametric Kruskal-Wallis test [36]. Differences with *p* < 0.05 were considered as statistically significant. All experiments were replicated at least three times.

## 3. Results

A total of 88 ovaries were included in the experiments. Eight hundred twenty-seven COCs were selected, of which 241 COCs were used as the control group and 591 COCs were exposed to BCB test. After BCB staining, 356 COCs (60.2%) were classified as BCB+ and 230 COCs (27.8%) as BCB−. Of these oocytes, 212 were used to evaluate the meiotic configuration, 147 for the evaluation of mitochondrial activity and ROS levels, and 468 for in vitro embryo production.

### 3.1. Effects of BCB Staining on Nuclear Maturation

Table 1 summarizes the results of the meiotic progression of the COCs stained with BCB before in vitro maturation.

No significant difference in nuclear maturation and degeneration rate were observed between oocytes exposed to BCB staining (BCB+ plus BCB−) and unexposed oocytes. The percentage of oocytes that progressed to metaphase II stage was similar between control and BCB+ groups and lower (*p* < 0.05) in BCB− oocytes compared to control and BCB+. No difference was found in the degeneration rate among groups. The percentage of BCB− oocytes at GV stage (27.8%) was higher (*p* < 0.05) than that of control (8.9%) and BCB+ (12.5%) oocytes. The rate of oocytes that resumed meiosis (GVBD and MI/TI stage) did not differ among groups.

### 3.2. Effects of BCB Staining on Mitochondria Activity and Reactive Oxygen Species (ROS) Levels

Mitochondria activity of BCB exposed (BCB+ plus BCB−) oocytes immediately after staining (*n* = 57; 0.95 ± 0.07 pixels\oocyte) and after in vitro maturation (*n* = 39; 0.96 ± 0.05 pixels\oocyte) was similar to the respective counterpart of control groups (*n* = 29; 1 ± 0.09 and *n* = 22; 1 ± 0.04 pixels\oocyte, *p* > 0.05; Figure 2A). Brilliant Cresyl Blue exposure did not affect intracellular ROS level in immature (*n* = 57; 1.18 ± 0.16 pixels\oocyte) and in in vitro matured (*n* = 39; 0.93 ± 0.04 pixels\oocyte) oocytes compared to control oocytes (*n* = 29; 1 ± 0.15 and *n* = 22; 1 ± 0.04 pixels\oocyte, respectively, *p* > 0.05; Figure 2B).

Mitochondria activity immediately after staining was not statistically different among the control (*n* = 29; 1 ± 0.09 pixels\oocyte), BCB+ (*n* = 34; 0.98 ± 0.11 pixels\oocyte) and BCB− (*n* = 23; 0.92 ± 0.10 pixels\oocyte) groups. After in vitro maturation, mitochondria activity was lower in the BCB− (*n* = 17; 0.73 ± 0.06 pixels\oocyte) group than in the BCB+ (*n* = 22; 0.98 ± 0.11 pixels\oocyte) and control (*n* = 22; 1 ± 0.04 pixels\oocyte) groups (*p* < 0.05, Figure 2A). Intracellular ROS levels in BCB− oocytes were higher immediately after staining (*n* = 23; 1.76 ± 0.21 pixels\oocyte) and lower after in vitro maturation (*n* = 17; 0.8 ± 0.05 pixels\oocyte) compared to controls (*n* = 29; 1 ± 0.18 and *n* = 22; 1 ± 0.04 pixels\oocyte, respectively) and BCB+ (*n* = 34; 0.89 ± 0.17 and *n* = 22; 1.02 ± 0.05 pixels\oocyte, respectively) oocytes (*p* < 0.05, Figure 2B).

### 3.3. Effect of BCB Selection on In Vitro Embryo Production

The results obtained after in vitro fertilization and embryo culture of the oocytes retrieved from the ovaries of anestrus cats selected by BCB staining are reported in Table 2. The cleavage rate was similar between BCB+ and control groups. BCB− oocytes cleaved at lower percentage (11.0%) compared to control (28.6%) and BCB+ (28.6%) oocytes (*p* < 0.05).

The percentage of blastocyst formation and hatching blastocysts (Figure 3A) was higher in the BCB+ group than in the control and BCB− groups. The number of total and TE cells of blastocysts developed from BCB+ oocytes was higher (*p* > 0.05) than in control and BCB− oocytes. BCB− group presented the lowest (*p* > 0.05) total and TE cell number (Figure 3B). No significant differences were found in ICM numbers among groups (Table 2).

## 4. Discussion

In domestic cats, the circannual variations in the oocyte quality is reflected by a reduced in vitro nuclear maturation and developmental competence during the non-breeding season [10,11,12]. Oocyte selection based on glucose-6-phosphate dehydrogenase (G6PDH) activity has been successfully used to differentiate between competent and incompetent oocytes in both adult and pre-pubertal animals [16,20,21,22,24,37,38,39].

Although it is generally demonstrated that BCB+ oocytes are more competent than BCB− ones, the lack of significant difference between blastocysts developed from BCB+ and unstained control oocytes observed in some studies [19,25,40], has led to doubts as to the real utility of the BCB test in IVEP technology [16,39].

In this research, we explored the effectiveness of BCB selection of high-quality immature oocytes retrieved from domestic cats during anestrous to ameliorate the outcome of in vitro embryo production. The main findings of our study proved that BCB stain before IVM allows identifying oocytes according to their meiotic and developmental competence.

More specifically, BCB+ oocytes developed to blastocyst stage, after IVF, at higher percentages than BCB− and unstained control oocytes. Furthermore, blastocysts produced from BCB+ oocytes had an enhanced quality as demonstrated by their increased capability to hatch and their higher cell number. Results also proved that the cell numbers of blastocysts deriving from BCB− oocytes were lower than those of control oocytes despite the rate of blastocyst development was similar between the two groups. This indicate that the low competence of the BCB− oocytes was reflected in the quality of the blastocysts produced. BCB− oocytes, therefore, represent a fraction of morphologically selected oocytes which would be likely to generate poor quality embryos.

In cattle [20,21], sheep [41], buffalo [42], and dromedary [43], BCB+ oocytes have yielded better results in terms of blastocysts produced after IVF compared to morphologically selected oocytes. Our findings confirmed the benefit of BCB classification of oocytes in addition to morphological appearance.

Our rate of blastocyst development (20%) from oocytes with low G6PDH activity was superior to that reported by other authors during the non-breeding season [10,12] and close to that achieved for IVM/IVF oocytes collected from cats during reproductive season [10,44].

Previously, Comizzoli et al. [12] demonstrated that supplementing the conventional IVM medium with FSH and antioxidant ascorbic acid or cysteine overcame poor in vitro nuclear and developmental competence of cat oocytes during non-breeding season. In our study, a conventional IVM medium was used, as our primary aim was to test the efficacy of the BCB selection method. Nevertheless, embryonic development of BCB+ oocytes was comparable to results achieved by the modified IVM medium in the study from Comizzoli et al. [12].

In an earlier research, the utility of BCB staining for pre-selection of domestic cat oocytes before IVM was tested [24]. In agreement with our results, this study reported a significant increase of nuclear maturation and embryonic development of BCB+ oocytes. However, the lack of unstained control oocytes in Jengenow’s study (2019) does not make it possible to definitely confirm the benefits of BCB pre-selection to ameliorate the outcome of IVEP [24].

It is difficult to compare the results of Jengenow et al. [24] and our results. The most important differences between the two studies include: (i) the time of the year when the oocytes were collected (July–November vs. over the year); (ii) BCB concentration and time of exposure (13 μM, 45 min, vs. 34 μM, 60 min), and (iii) the stage of embryo development which was assessed after IVF (blastocyst vs. morula stage).

Several evidences suggested a correlation between the concentration of BCB and the accuracy of the selection method [19,22,45], thus low concentrations of the dye (13 μM and 26 μM) are usually used.

In our study, the selection of cat oocytes with 13 μM BCB for 45 min increased blastocyst rate from 7.3% (BCB−) to 20.0% (BCB+).

Our preliminary experiments have been addressed to examine whether BCB staining can have potentially damaging effects on bioenergetic/oxidative status of oocytes. Recently, adverse effects due to BCB exposure of the oocyte have been described. In particular, Santos et al. [29] proved that BCB staining altered the functioning of the mitochondria and caused an overproduction of ROS in immature and IVM pig oocytes. In contrast, our findings established that BCB staining of cat oocytes did not affect mitochondrial activity and ROS levels, neither immediately after BCB exposure nor after IVM. The differences between the findings of Santos et al. [29] and our findings may be related to the length of oocyte exposure to BCB (60′ vs. 45′) and to the different sensitivity of the oocytes to the dye in relation to the age of the donor (pre-pubertal sows vs. adult queens) and the species.

Mitochondria are commonly studied as a marker of oocyte quality and health, their proper functioning is crucial for providing the energy necessary for oocyte maturation and for embryo development, as well as for regulating redox homeostasis [46]. In cattle, Torner et al. [47] found that BCB− immature oocytes had an increased mitochondrial activity compared to BCB+ oocytes. In contrast, mitochondrial activity was lower in pig immature BCB– oocytes compared to BCB+, but no differences have been recorded between the two classes of oocytes after IVM [48].

We found that the mitochondria activity of BCB+ and BCB− immature cat oocytes did not differ, while BCB− oocytes produced ROS at higher levels than BCB+ oocytes.

Physiological levels of ROS modulate oocyte functions [49], while an excessive production leads to oxidative stress and may cause a progressive loss of oocyte developmental competence [50,51]. Our results suggest that BCB− oocytes may have an inadequate antioxidant defense system and/or most likely originate from a sub-optimal follicular environment responsible for greater oxidative stress. Therefore, an impaired potential for glutathione (GSH) synthesis was observed in prepubertal low quality oocytes compared to adult ovulated oocytes in mice [52]. Differences of antioxidant enzymes between immature BCB+ and BCB− oocytes will be investigated.

Our data also showed that after in vitro maturation BCB− oocytes had lower mitochondria activity and ROS levels compared to BCB+ oocytes. This would indicate a perturbed metabolic activity of BCB− oocytes likely to be reflected by a decrease of cleavage and embryonic development after IVF.

In agreement with our results, Català et al. [22] reported that while no differences in mitochondrial activity were recorded between BCB+ and BCB− sheep immature oocytes, BCB− oocytes displayed a lower mitochondrial activity after IVM.

Collectively, our results provide preliminary indications for the selection of best quality oocytes and for achieving acceptable cat embryos rates throughout the year.

## 5. Conclusions

In conclusion, the findings of our study indicate that the selection of oocytes by combined morphological criteria and BCB staining can be an efficient method for classifying oocytes derived from anestrus cats. Oocytes with low G6PDH activity had an enhanced ability to develop to blastocyst stage after IVM/IVF. Moreover, blastocysts developed from BCB+ oocytes showed higher quality as defined by an increased proportion of hatching blastocyst and higher cell number. The BCB test could provide a useful tool to support the selection of good quality oocytes and to improve IVEP outcome from felines of high genetic value that died during the non-breeding season. The intrinsic molecular and subcellular features associated with developmental potential of cat oocytes when selected on the basis of G6PDH activity are currently under investigation in our laboratory.

## Figures and Tables

**Figure 1 animals-10-01496-f001:**
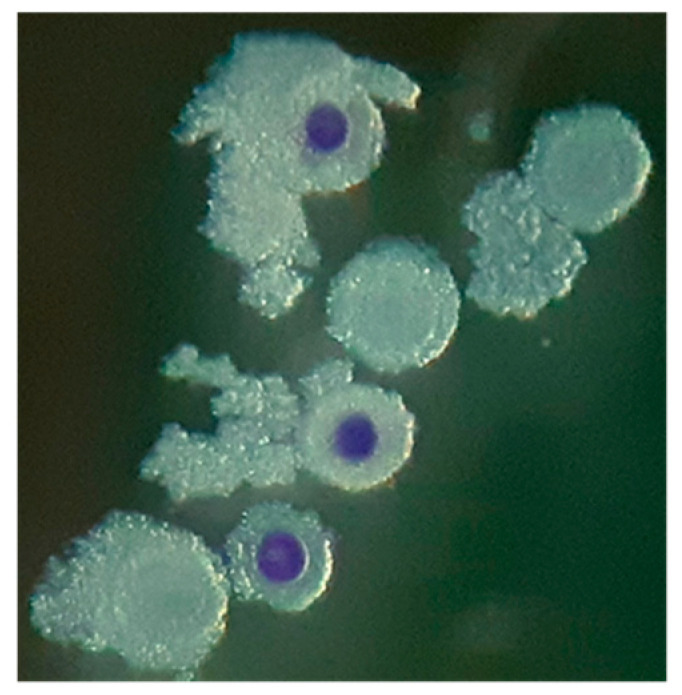
High quality cumulus oocyte complexes, collected from ovaries of anestrus cats, stained by Brilliant Cresyl Blue (BCB).

**Figure 2 animals-10-01496-f002:**
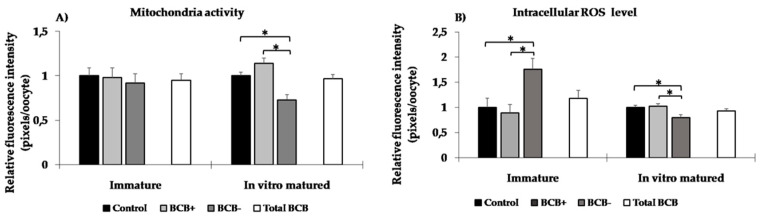
Mitochondria activity (**A**) and intracellular reactive oxygen species (ROS) level (**B**) in domestic cat oocytes immediately after Brilliant Cresyl Blue (BCB) staining and after in vitro maturation. Values are expressed as mean ± standard error of the mean (SEM). Immature: germinal vesicle; in vitro matured: metaphase II; total BCB: BCB+ plus BCB−. Total BCB group was compared to control group. Symbol * indicate significant difference (*p* < 0.05).

**Figure 3 animals-10-01496-f003:**
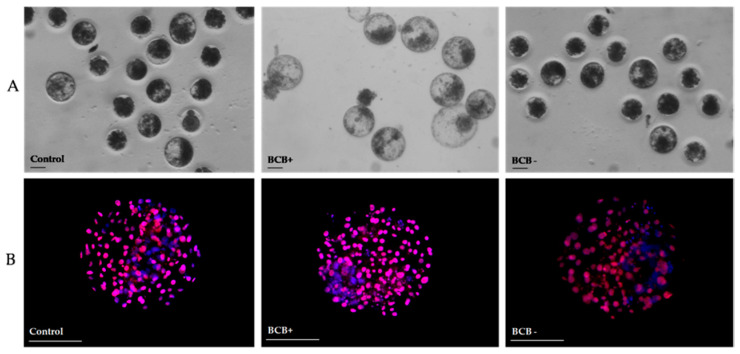
Representative phase-contrast light micrographs of blastocyst produced in vitro from cat oocytes selected by Brilliant Cresyl Blue (BCB) staining (**A**) and epifluorescence micrographs of cat blastocysts after differential staining (**B**). Inner cell mass appears in blue (Hoechst) and trophectoderm in red (propidium iodide). Scale bar = 100 μM.

**Table 1 animals-10-01496-t001:** Meiotic progression of cat oocytes obtained from ovaries of anestrus cats selected by Brilliant Cresyl Blue (BCB).

Group	Total Oocytes *n*	Nuclear Chromatin Configuration *n* (%)				Degenerated *n* (%)
		GV	GVBD	MI\TI	MII	
Control	78	7 (8.9) ^a^	6 (7.7)	6 (7.7)	50 (64.1) ^a^	9 (11.5)
BCB+	80	10 (12.5) ^a^	5 (6.2)	4 (5.0)	54 (67.5) ^a^	7 (8.7)
BCB−	54	15 (27.8) ^b^	6 (11.1)	2 (3.7)	20 (37.0) ^b^	11 (20.4)
Total BCB	134	25 (18.6) ^ab^	11 (8.2)	6 (4.5)	74 (55.2) ^a^	18 (13.4)

^ab^ Different superscript letters within the same column indicate significant difference (*p* < 0.05). GV: Germinal vesicle; GVBD: Germinal vesicle breakdown; MI: Metaphase I; TI: Telophase I; MII: Metaphase II.

**Table 2 animals-10-01496-t002:** Embryonic development after in vitro maturation and fertilization of domestic cat oocytes obtained from ovaries of anestrus cats by Brilliant Cresyl Blue (BCB) staining.

Group	No. Oocytes	Cleaved (%)	Blastocyst (%/n)	Hatched Blastocysts (%/n)	Cell Number (Mean ± SEM)		
					Total	TE	ICM
Control	112	32 (28.6) ^a^	12 (10.7) ^a^	4 (3.6) ^a^	197.1 ± 8,7 ^a^	152.6 ± 8.5 ^a^	44.4 ± 2.6
BCB+	220	63 (28.6) ^a^	44 (20.0) ^b^	26 (11.8) ^b^	237.8 ± 12.5 ^b^	185.8 ± 9.5 ^b^	51.7 ± 3.8
BCB−	136	15(11.0) ^b^	10 (7.3) ^a^	2 (1.5) ^a^	130 ± 13.7 ^c^	104.4 ± 13.6 ^c^	37.7 ± 5.2

^abc^ Different superscript letters within the same column indicate significant difference (*p* < 0.05). ICM, inner cell mass; TE, trophectoderm.

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
