# Peer review of "Selection of Immature Cat Oocytes with Brilliant Cresyl Blue Stain Improves In Vitro Embryo Production during Non-Breeding Season"

_animals, 2020, doi:10.3390/ani10091496_

Round 1

Reviewer 1 Report

In this study, AA aimed to evaluate the effect of Brilliant Cresyl Blue (BCB) staining on cat oocyte development during the non-breeding season. Domestic queens in anestrus were used to collect COCs, that were stained for BCB or not and subjected to IVM. Mitochondrial activity and ROS were measured before and after IVM. 

Then, in vitro matured oocytes were fertilized up to follow embryonic cleavage, blastocyst formation and hatching; the blastocyst cell number was recorded.

Results showed that BCB+ oocytes developed to blastocyst stage, after IVF, at higher percentages than those of BCB- and unstained control oocytes. Furthermore, blastocysts produced from BCB+ oocytes had an enhanced quality as demonstrated by their increased capability to hatch and their higher cell number. 

The study, of interest for those working in the field, is clear and quite well written. The study design is appropriate and results are novel, and appropriately discussed at the light of relevant literature. However, before possible acceptance, some revision is required.

The approval of IRB should be inserted if needed.

AA reports the total number of COCs used for BCB staining; it is, however, suggested to indicate the number of COC used for mitochondrial activity/ROS and that for IVC.

The rationale of split data on mitochondrial activity and ROS in two figures, i.e. C vs BCB and C vs BCB+ vs BCB-, it is unclear. Could you, please, strengthen the importance of providing both figures instead only Fig. 3?

Why AA did not test the mitochondrial activity and ROS also at the end of embryonic development? data could probably reinforce the positive results on the efficacy of BCB.

A careful proof-read spell check is needed to eliminate grammatical errors.

Author Response

Dear Editor,

Thank you for considering our manuscript for publication in Animals. We are grateful to you and to the reviewers for the constructive suggestions provided.

We have revised the manuscript in view of reviewer’s comments as outlined in detail in the cover letter below.

We marked the revisions in red (referee 1), and blue (referee 2).

Therefore, as Editor suggested, we checked the manuscript by a native English speaker to improve the scientific English.

Best Regards

Anna Rita Piras

Response to Reviewer 1 Comments

We would like to thank the reviewer for his/her careful and helpful revision and constructive suggestions, which gave us the opportunity to improve the quality of this manuscript.

We modified the manuscript according to the suggested revisions and we marked changes in red.

Response to specific reviewer’s comments:

Point 1: -The approval of IRB should be inserted if needed.Ethical statement

Response 1: Live animals raising ethical concerns were not used in this study. The samples (ovaries) were collected during routine ovariectomies at the Veterinary Teaching Hospital of the University, following the basic criteria of good veterinary surgery practices. We added this information in Materials and Methods (2.1 Animal selection) line 108-110.

Point 2: -AA reports the total number of COCs used for BCB staining; it is, however, suggested to indicate the number of COC used for mitochondrial activity/ROS and that for IVC.

Response 2: According to the reviewer’s suggestion we added the total number of COCs used for meiotic configuration, mitochondrial activity/ROS and that for IVEP (line 240-242). In addition, numbers of oocytes belonging to the various groups used for each experiment are indicated in Table 1 (meiotic progression), in Table 2 (embryonic development) and in the section 3.2 (Mitochondrial activity and ROS levels).

Point 3: -The rationale of split data on mitochondrial activity and ROS in two figures, i.e. C vs BCB and C vs BCB+ vs BCB-, it is unclear. Could you, please, strengthen the importance of providing both figures instead only Fig. 3?

Response 3:We agree with reviewer comments regarding the two figures.

We inserted two distinct figures to highlight two different aspects.

  1. In figure 2 the mitochondrial activity and the ROS levels of the oocytes exposed or not to the BCB staining were compared, in order to evaluate any negative effects of the dye exposure, (control vs BCB-treated oocytes).
  2. In figure 3, we compared the mitochondrial activity and the levels of ROS among individual groups (control vs BCB+ vs BCB-) in order to identify possible differences between the BCB + and BCB- oocytes as marker of oocyte quality.

However, according to reviewer’s suggestion and to avoid misunderstanding we combined figures 2 and 3 in a single figure (Figure 2).

Point 4: -Why AA did not test the mitochondrial activity and ROS also at the end of embryonic development? data could probably reinforce the positive results on the efficacy of BCB.

Response 4: We agree with the reviewer that it could be relevant to assess the mitochondrial activity and ROS also at the end of embryonic developmentto expand evaluation on embryo quality and reinforce the positive results on the efficacy of BCB. In this preliminary explorative study, taking into account the number of samples, first we have chosen to subject the blastocysts produced to differential staining in order to assess their quality in terms of total ICM and TE cell numbers.

Further investigations, including assessment of mitochondria activity and ROS levels, could give more insights regarding the quality of blastocyst produced by BCB selection of oocytes.

Point 5: A careful proof-read spell check is needed to eliminate grammatical errors.

Response 5:According to the reviewer’s suggestionwe checked the manuscript by a native English speaker to eliminate grammatical errors.

Reviewer 2 Report

General comments:
The authors used a stain test, Brilliant 28 Cresyl Blue (BCB) staining, to determine if cat oocytes maturation can be predicted better by using this sorting test. They also checked toxicity of the stain by measuring mitochondrial activity and levels of reactive Oxygen Species.
They concluded that the BSB stain can be used to select oocytes that will mature and survive (up to 7 days) in significantly higher numbers than the control cells (selected with no sorting) while using in-vitro fertilization techniques.
The study is legitimate and may help better select oocytes for in vitro fertilization in Felids, but aiming to be able to breed during the quiet session (when the oocytes are in anestrous stage and therefore oocytes waves are quietness) might be too ambitious.
The authors should review their terminology about oocytes and follicle stages – the oocyte would burst in ovulation, but is present in all the developmental stages of follicles within the ovary (i.e. it is part of the primary, secondary and tertiary follicles).
Further explaining the rational of your study will enhance your argument - why it is needed to use follicles of anoestrus animals and not wait to the natural breeding season?
General remark on discussion structure – some arguments present as literature review follows by your finding. I suggest to improve the flow by discussing your findings in comparison to the related studies mentioned.
I would also suggest adding a discussion point on why the number of BCB- cells were lower than the controls.
Specific comments:
Line 25 – typo – should be IVEP
Missing space between the end of one sentence and the beginning of the following sentence in many occasions throughout the manuscript. E.g., line 28 between “Season.” And “This”, line 36 between “recorded.” And “The”, Line 37 between “oocytes.” And “BCB+”. Please correct throughout the manuscript.
Line 46 “in vitro” – italics? (check style guidance)
“In the temperate zone…” – the cat is depends on length of day and not temperature, would it be better to use northern versus southern hemispheres instead of climate type?
Line – “blastocyst stage of IVM\IVF cat oocytes” – the blastocyte stage happens after fertilization and consider as a stage of the embrio, not of the oocyte (also correct in the abstract/summary)
Line 68 – as this is not a reproduction journal, suggest to spell the full name of FSH (here) and LH (Line 167).
Line 73 – “Oocytes that have finished their growth phase (fully grown or more competent oocytes” – there are accepted terms to maturation of different follicle stages of in the ovary – did you mean tertiary follicle, i.e. follicles that are just prior to ovulation?
Line 113 – “dark cytoplasm” – did you mean “blue” or ”dark blue colour”?
Line 121 – “according to their cytoplasm coloration” – It seems to me that the oocyte cytoplasm is stained with blue colour but not the granulosa cells surrounding it? (“their” refers to the whole complex)
Line 118 – “0.5 g/LBSA “ – separate “0.5 g/L” from “BSA” & abbreviations for BSA should be presented here and not in line 142.
Line 151 – “Vaseline®” – remove “®” symbol.
Line 160 – “LAS lite 170 Image 160 software” version and company that produce this software?
Line 165 – should it be “Hepes-buffered TCM 199” (not just TCM 199?)
Line 206 –“ followed by Bonferroni’s” – do you mean “Bonferroni correction”?
Lines 204 to 210 – consider adding references to the less-common statistical tests (e.g. Kruskal-Wallis test.)
Line 222 – “The percentage of BCB- oocytes at GV stage was higher than that of control and BCB+ oocytes” – insert relevant values & was this significantly higher?
Line 261 – “cleavage rate was comparable” – did you mean similar? (you compared all of your values)
Line 262 – specify the percentage value.
Line 280 – “sane” – a typo?
Line 312 – “in Jengenow’s study” – Add year of study in brackets
Line 326 – “observed in pig oocytes” – were these oocytes taken during anestrus as well?
Tables:
Table 1, 2, 3 footnote - a vs b Values with different superscript letters within a column differ significantly” – it is not clear which comparison each letter represents? E.g., I think you meant that the value marked with a “b” in the left column is different from both columns above it?
The second right column of Table 2 has the same data as in the second left column of Table 3 – combine?
Figures:
Figures 2 and 3 can be combined to one figure (with adding of the total BCB from Figure 2 as additional column in Figure 3)
Revise the way statistical significant is presented in your Figure 3. Better to use a star for significant value, denoting with line to which comparison it belongs.

Author Response

Dear Editor,

Thank you for considering our manuscript for publication in Animals. We are grateful to you and to the reviewers for the constructive suggestions provided.

We have revised the manuscript in view of reviewer’s comments as outlined in detail in the cover letter below.

We marked the revisions in red (referee 1), and blue (referee 2).

Therefore, as Editor suggested, we checked the manuscript by a native English speaker to improve the scientific English.

Best Regards

Anna Rita Piras

Response to Reviewer 2 Comments

We would like to thank the reviewer for his/her careful and helpful revision and constructive suggestions, which gave us the opportunity to improve the quality of this manuscript.

We modified the manuscript according to the suggested revisions and we marked changes in blue

Response to specific reviewer’s comments:

Point 1:-The authors should review their terminology about oocytes and follicle stages – the oocyte would burst in ovulation, but is present in all the developmental stages of follicles within the ovary (i.e. it is part of the primary, secondary and tertiary follicles).

Response 1:According to the reviewer’s indication we specified that the cumulus-oocyte complexes used for the study were recovered from antral follicles of ovaries during anestrus period (line 78 and 117).

Point 2:-Further explaining the rational of your study will enhance your argument - why it is needed to use follicles of anoestrus animals and not wait to the natural breeding season?

Response 2:We would like to thank the reviewer for this insightful comment.

We better explain the rationale of our experiment at the end of the introduction.

Line 91-97: “This study was conceived to test whether the selection of immature oocytes retrieved from domestic cat ovaries during non-breeding season, by a combination of COC morphology and staining with BCB, would be helpful in selecting developmentally competent oocytes, and thus would enhancing in vitro fertilization outcome. The possibility of classifying oocytes according to their developmental potential is particularly important to understand oocyte biology and might help to reduce negative impact of season on in vitro embryo production especially in the case of wild feline that die unexpectedly during anestrus period.

This concept had already been reported in the conclusions.

Point 3:-General remark on discussion structure – some arguments present as literature review follows by your finding. I suggest to improve the flow by discussing your findings in comparison to the related studies mentioned.

Response 3:According to the reviewer suggestion we tried to improve the discussion.

Point 4: -I would also suggest adding a discussion point on why the number of BCB- cells were lower than the controls.

Response 4:We would like to thank the reviewer for this suggestion.

We added a discussion point about the difference in the blastocysts cells number among BCB- and control groups

Line 323-328: “Results also proved that the cell numbers of blastocysts deriving from BCB- oocytes were lower than those of control oocytes despite the rate of blastocyst development was similar between the two groups. This indicate that the low competence of the BCB- oocytes was reflected in the quality of the blastocysts produced. BCB- oocytes, therefore, represent a fraction of morphologically selected oocytes which would be likely to generate poor quality embryos”.

Specific comments:

Point 5: -Line 25 – typo – should be IVEP

Response 5:Line 26: We changed IVPE with IVEP

Point 6: -Missing space between the end of one sentence and the beginning of the following sentence in many occasions throughout the manuscript. E.g., line 28 between “Season.” And “This”, line 36 between “recorded.” And “The”, Line 37 between “oocytes.” And “BCB+”. Please correct throughout the manuscript.

Response 6:We wish to thank the reviewer for this indication. We added space accordingly.

Point 7: -Line 46 “in vitro” – italics? (check style guidance)

Response 7:According to the guidelines for authors “in vitro” should not be written in italics

Point 8: -“In the temperate zone…” – the cat is depends on length of day and not temperature, would it be better to use northern versus southern hemispheres instead of climate type?

Response 8:We agree with reviewer comment and we corrected “in the temperate zones” with “in the northern hemisphere” (line 59).

Point 9: -Line 58 – “blastocyst stage of IVM\IVF cat oocytes” – the blastocyst stage happens after fertilization and consider as a stage of the embryo, not of the oocyte (also correct in the abstract/summary)

Response 9: We followed the advice of the reviewer and we corrected “blastocyst stage of IVM\IVF cat oocytes”

Line 62-64: the sentence “The in vitro developmental competence up to the blastocyst stage of IVM\IVFcat oocytes during the non-breeding season was significantly reduced compared to that of oocytes recovered during the reproductive season” was changed with“In vitro embryo development up to the blastocyst stage after IVM/IVF of domestic cat oocytes is significantly reduced during the non-breeding season compared to the reproductive season”.

We also modified summary (line 22).

Line 22: The sentence: “The yield of the blastocysts and the quality of the embryos (hatching rate and blastocyst cell numbers) were higher in the oocytes defined as better quality (BCB +)”was changed with“Blastocysts yield and quality of the embryos (hatching rate and blastocyst cell numbers) were higher after IVM/IVF in oocytes defined as BCB+ (colored cytoplasm) based on BCB test than in oocytes only morphologically selected”.

And abstract

Line 28: “In domestic cats, oocytes ability to develop to the stage of blastocyst in vitrodecreases during the non-breeding season” was changed with

“In domestic cats, the maturation, fertilization and development potential in vitro decreases during the non-breeding season”.

Point 10:-Line 68 – as this is not a reproduction journal, suggest to spell the full name of FSH (here) and LH (Line 167).

Response 10:The full name of the hormones FSH (Line 73) and LH (Line 183) was specified.

Point 11: -Line 73 – “Oocytes that have finished their growth phase (fully grown or more competent oocytes” – there are accepted terms to maturation of different follicle stages of in the ovary – did you mean tertiary follicle, i.e. follicles that are just prior to ovulation?

Response 11:We agree with referee comment. We better explained this point. As we reported in material and methods oocytes were recovered from antral follicles (line 117).

To avoid confusion, we modified Line 76-78: “Brilliant Cresyl Blue (BCB) staining is a non-invasive method for identifying oocyte quality that has been used in different animal species [16], including human [17]. BCB test relies on measurement of the glucose-6-phosphate (G6PDH) enzyme activity that converts the BCB stain from blue to colorless (BCB). Oocytes that have finished their growth phase (fully grown or more competent oocytes) have low levels of G6PDH and will exhibit blue colored cytoplasm (BCB+), whereas growing or less competent oocytes have a high levels of G6PDH and reduce the dye resulting in a colorless cytoplasm (BCB-) [18,19] “ as follows: “Brilliant Cresyl Blue (BCB) staining is a non-invasive method that has been used in different animal species [16], including human [17] for identifying the quality of oocytes retrieved from antral follicles.

Point 12: -Line 113 – “dark cytoplasm” – did you mean “blue” or ”dark blue colour”?

Response 12:In the line 122 the term “dark cytoplasm” refers to the morphological classification of cat oocytes (before BCB staining ) based on the selection criteria of Wood and Wildt 1997.

Point 13: -Line 121 – “according to their cytoplasm coloration” – It seems to me that the oocyte cytoplasm is stained with blue colour but not the granulosa cells surrounding it? (“their” refers to the whole complex).

Response 13:In order to indicate that the oocyte cytoplasm is stained with blue color but not the granulosa cells we replaced “their cytoplasm” with “oocytes cytoplasm” (Line 131).

Point 14: -Line 118 – “0.5 g/LBSA “ – separate “0.5 g/L” from “BSA” & abbreviations for BSA should be presented here and not in line 142.

Response 14:Line 128: “0.5 g/L” has been separated from “BSA” and the BSA full name has been reported.

Point 15: -Line 151 – “Vaseline®” – remove “®” symbol.

Response 15:Line 165: the symbol “®” has been removed.

Point 16: -Line 160 – “LAS lite 170 Image software” version and company that produce this software?

Response 16:Line 175: we added the name of the company that produce the software used for the fluorescence quantification.

Point 17: -Line 165 – should it be “Hepes-buffered TCM 199” (not just TCM 199?)

Response 17:For in vitro maturation we used TCM 199 and not Hepes-buffered TCM 199. Hepes-buffered TCM 199 is indicated for procedures performed at room atmosphere. TCM 199 buffered with sodium bicarbonate is generally used to incubate oocytes in the atmosphere with 5% CO2.

We specified that NaHCO3-TCM 199 has been used for IVM (line182).

Point 18: -Line 206 –“ followed by Bonferroni’s” – do you mean “Bonferroni correction”?

Response 18:Line 230: “Bonferroni’s” has been substituted by “Bonferroni correction”

Point 19:-Lines 204 to 210 – consider adding references to the less-common statistical tests (e.g. Kruskal-Wallis test.)

Response 19:Line 227 to 232 According to the reviewer’s suggestion we added the references for Shapiro-Wilk test (Mishra et al. 2019) and for Kruskal-Wallis test (Bewick et al. 2004).

Point 20: -Line 222 – “The percentage of BCB- oocytes at GV stage was higher than that of control and BCB+ oocytes” – insert relevant values & was this significantly higher?

Response 20:Line 251-252: According to the reviewer’s suggestion we added the relevant values and the P value.

Point 21: -Line 261 – “cleavage rate was comparable” – did you mean similar? (you compared all of your values).

Response 21:Line 285: “cleavage rate was comparable” was substituted by “cleavage rate was similar”.

Point 22: -Line 262 – specify the percentage value.

Response 22:Line 287: The percentage values of the cleaved embryos were reported in the text.

Point 23: -Line 280 – “sane” – a typo?

Response 23:Line 296 “sane” has been changed in “same”, thanks for finding this typo.

Point 24: -Line 312 – “in Jengenow’s study” – Add year of study in brackets

Response 24:Line 346: the year of the study has been added.

Point 25: -Line 326 – “observed in pig oocytes” – were these oocytes taken during anestrus as well?

Response 25:In the study of Santos et al. the negative effect of BCB staining on mitochondrial activity and ROS levels was investigated in oocytes retrieved from ovaries of pre-pubertal gilts (6 months old).

Tables:
Point 26: -Table 1, 2, 3 footnote - a vs b Values with different superscript letters within a column differ significantly” – it is not clear whichcomparison each letter represents? E.g., I think you meant that the value marked with a “b” in the left column is different from both columns above it?

values with different letters differ significantly

Response 26:According to reviewer suggestion footnotes of Tables 1and 2 have been corrected with:

“Different superscript letters within the same column indicate significant difference”.

Point 27: -The second right column of Table 2 has the same data as in the second left column of Table 3 – combine?

Response 27:According to reviewer suggestion Table 2 and Table 3 were combined in a single table (Table 2).

Figures:
Point 28: - Figures 2 and 3 can be combined to one figure (with adding of the total BCB from Figure 2 as additional column in Figure 3)

Response 28:According to the reviewer’s suggestion we combined the Figure 2 and 3.

Point 29: -Revise the way statistical significant is presented in your Figure 3. Better to use a star for significant value, denoting with line to which comparison it belongs.

Response 29:Statistical difference representation in the Figure 2 (ex Figure 2 and 3) were modified as reviewer’s request.

Round 2

Reviewer 1 Report

The AA revised the ms according to the received suggestions. The ms is now suitable for pubblication.